# Perineural Invasion in Pancreatic Ductal Adenocarcinoma: From Molecules towards Drugs of Clinical Relevance

**DOI:** 10.3390/cancers14235793

**Published:** 2022-11-24

**Authors:** Federico Selvaggi, Eugenia Melchiorre, Ilaria Casari, Sebastiano Cinalli, Massimiliano Cinalli, Gitana Maria Aceto, Roberto Cotellese, Ingrid Garajova, Marco Falasca

**Affiliations:** 1Unit of Surgery, Renzetti Hospital, ASL2 Lanciano-Vasto-Chieti, 66034 Lanciano, Italy; 2Department of Medical, Oral and Biotechnological Sciences, “G. d’Annunzio” University, Chieti-Pescara, Via dei Vestini 31, 66100 Chieti, Italy; 3Metabolic Signalling Group, Curtin Health Innovation Research Institute, Curtin Medical School, Curtin University, Perth, WA 6102, Australia; 4Department of Pathology, Sondrio Hospital, 23100 Sondrio, Italy; 5Villa Serena Foundation for Research, 65013 Pescara, Italy; 6Medical Oncology Unit, University Hospital of Parma, Via Gramsci 14, 43126 Parma, Italy

**Keywords:** pancreatic ductal adenocarcinoma, perineural invasion, tumour microenvironment, neuropathic pain, pancreas

## Abstract

**Simple Summary:**

A few types of cancers are currently as challenging and difficult to defeat as pancreatic ductal adenocarcinoma. Several reasons contribute to the complexity of this disease and are extensively studied in the attempt to beat this unassailable condition. Among those factors, the invasion of nerves by cancer cells, or perineural invasion, has been discovered to be a common feature of this cancer helping the tumour in its progression, facilitating relapses and causing considerable pain for patients. For these reasons, more effective therapies directed at inhibiting nerve invasion promoted by pancreatic ductal adenocarcinoma are strongly advocated. This review discusses the current understanding of perineural invasion in pancreatic ductal adenocarcinoma and the state of the art regarding pharmacological progress in this field.

**Abstract:**

Pancreatic ductal adenocarcinoma is one of the most threatening solid malignancies. Molecular and cellular mediators that activate paracrine signalling also regulate the dynamic interaction between pancreatic cancer cells and nerves. This reciprocal interface leads to perineural invasion (PNI), defined as the ability of cancer cells to invade nerves, similar to vascular and lymphatic metastatic cascade. Targeting PNI in pancreatic cancer might help ameliorate prognosis and pain relief. In this review, the modern knowledge of PNI in pancreatic cancer has been analysed and critically presented. We focused on molecular pathways promoting cancer progression, with particular emphasis on neuropathic pain generation, and we reviewed the current knowledge of pharmacological inhibitors of the PNI axis. PNI represents a common hallmark of PDAC and correlates with recurrence, poor prognosis and pain in pancreatic cancer patients. The interaction among pancreatic cancer cells, immune cells and nerves is biologically relevant in each stage of the disease and stimulates great interest, but the real impact of the administration of novel agents in clinical practice is limited. It is still early days for PNI-targeted treatments, and further advanced studies are needed to understand whether they could be effective tools in the clinical setting.

## 1. Introduction

Pancreatic ductal adenocarcinoma (PDAC) is a dismal disease characterized by poor prognosis, recurrence and resistance to chemotherapy [1,2]. PDAC is the second most frequent cancer of the gastrointestinal tract, the third principal cause of cancer overall and the fourth major fatal illness in the United States. Curative surgery, defined as R0 resection and adjuvant chemotherapy, is still currently the canonical regimen for patients with resectable, non-metastatic PDAC [3]. In advanced stages, when in the presence of local or metastatic spread, a combination of gemcitabine and Nab-paclitaxel or mFOLFIRINOX is the usual treatment [4]. PDAC is asymptomatic in the initial stages of the disease with high heterogeneity. Unfortunately, this malignancy is characterized by chemoresistance and lack of response to radiation treatments. Moreover, the few available chemotherapy treatments are frequently associated with adverse effects. The first-line treatment for PDAC has been based on antimetabolite gemcitabine since 1997 [2]. In the PRODIGE 4/ACCORD 11 trial, patients were treated randomly with mFOLFIRINOX (5-fluorouracil, oxaliplatin, irinotecan) or gemcitabine alone [5]. FOLFIRINOX improves the global health status and the emotional functioning of metastatic PDAC patients. According to the National Comprehensive Cancer Network guidelines, 5-fluorouracil/leucovorin, oxaliplatin, irinotecan (FOLFIRINOX) and gemcitabine, plus albumin-bound paclitaxel (nabPTX+GEM), are the favoured first-line treatments [6]. PDAC development and progression appear to be under complex neural influences where sensory and sympathetic nerves stimulate tumour growth, while parasympathetic nerves inhibit tumorigenesis through cholinergic signalling [7,8,9]. Nerves are components of the tumour microenvironment and participate in all stages of cancer, even in precursor lesions such as pancreatic intraepithelial neoplasia (PanIN) [10,11]. Perineural invasion (PNI) is an omnipresent feature of PDAC, which, although not yet thoroughly understood, is known to have a negative influence on prognosis [12]. This review will examine the clinical significance of perineural invasion and focus on signalling pathways and therapeutic opportunities that may benefit PDAC patients.

## 2. Perineural Invasion

When cancer cells are found in a minimum of 33% of the nerves and have infiltrated the epineural, perineural and endoneurial layers of the neural sheath, a PNI is identified. Historically, neural cancer cell invasion has been recognized and defined at the beginning of the 19th century [13]. Cancer cell invasion of nerves is common in a number of cancers, and in PDAC, the prevalence of PNI is high, reaching up to 100% [7,8,11,13]. PNI comprises manifestations of “neural invasion”, another definition for tumour cell invasion of the nerves greater than 33% [11,14]. Patients that survived longer than 3 years after surgery presented no extra-pancreatic nerve invasion [7]. Pathogenesis of PNI is based on “the low-resistance channel”, a hypothesis in which cancer cells spread passively along nerves, and “the mutual attraction” theory, in which nerves, cancer cells and stromal cells interact actively [15]. However, modern studies consider PNI to be an active invasion rather than passive cancer cell diffusion [10,11] (Figure 1).

### 2.1. PNI Overview

A Medline search using the term “perineural invasion in human cancer” shows 4871 results (November 2022). When the search is focused on the specific term “perineural invasion in human pancreatic cancer,” the final number of published articles is 714. PNI is a common characteristic of several human cancers. For example, it is frequently found in squamous cell carcinoma, the recurrent cancer of the head and neck. Molecular markers are NGF, BDNF, GDNF, Semaphorin, Galanin, CX3CL1/CX3CR1, Galectin 1, Cytokine A, NCAM, ICAM-5, IMP3 and BAG1. For nose and paranasal sinuses cancer, the PNI detection is 25 to 46.2%. For larynx and hypopharynx cancer, it is 46%. For the oral cavity and oropharynx cancer, the PNI is 26.3 to 72.1%. For the tongue and/or floor of the mouth cancer, the PNI detection rate is 6 to 71% [16]. The occurrence of reported PNI in oral squamous cell carcinoma is very inconsistent; it ranges from a 2% low frequency to a high of 82%, especially when using neural staining [17]. In addition, patients with invasive breast cancer who underwent surgery and presented PNI may have an increased probability of loco regional recurrence. Precisely, 1384 (15.6%) out of 8864 of those tumours were found to be accompanied by PNI. After 6.3 years, patients with PNI presented 428 loco regional recurrence events yielding a 7-year LRR of 7.1% (95% CI 5.5–9.1), while subjects clean of PNI had only 4.7% (95% CI 4.2–5.3; *p* = 0.01) [18]. When colon cancer is considered, of 21,488 patients evaluated, 55.2% had T3 disease (n D 11,852), 23.1% had T2 (n D 4971), 14.4% had T1 (n D 3088) and 7.3% had T4 disease (n D 1577); 4.6% (n D 987) had PNI. Concerning colon cancer, PNI is an independent poor prognostic factor for stage T3 and stage T4 tumours. Furthermore, patients with T3–4N0 colon cancer and PNI treated with adjuvant chemotherapy were found to have an improved chance of survival [19]. The biology of PNI is correlated with prostate cancer’s progression and defines the lethal phenotype [20].

### 2.2. PNI in Pancreatic Cancer

Although the molecular mechanisms of PNI are common features in different human cancers, the prevalence of PNI in PDAC surpasses any other solid malignancy. Surprisingly, PNI has a prevalence that reaches 100% in PDAC. The fact that the pancreas is found in close proximity to several neural plexuses helps to understand why this organ is particularly innervated. In addition, PNI’s pattern is related to the site of the tumour. Schematically, for tumours of the pancreatic head, the cancer cells spread towards the celiac plexus and ganglion along the pancreaticus capitalis I plexus. Alternatively, in the presence of uncinated process tumours, the metastatic cells move in the direction of the superior mesenteric plexus along the inferior pancreaticoduodenal artery plexus, while the pancreatic body and tail cancer spreads all the way to the splenic and the celiac plexus. The specificity of PNI in PDAC should be seen according to the functional role of the pancreas, with its endocrine and exocrine molecular mechanisms, its anatomical location as discussed and its relationship with surrounding organs. These aspects, together with the strong neurotropism of PDAC cells in nerves, represent important peculiarities. The pancreas is a retroperitoneal organ surrounded by the celiac plexus, the dorsal hepatic plexus and the plexus around the superior mesentery artery. Typically, nerves are located both in the periphery and in the internal part of the tumours [9]. As a result of PDAC cell invasion, neural damage with perineurium disruption and nerve distortion with oedema of axons is observed [13].

A novel standardized scoring system is introduced to differentiate PNI and endoneural invasion (ENI). Although cancer cells commonly grow in the perineural space (PNI), the more aggressive pattern is characterized by the direct and deeper invasion of nerves in the intrafascicular connective tissue, called endoneurium. Patients with ENI have more intense pain compared with those affected by PNI [13]. PDAC with a high probability to develop PNI (70–100%) has a negative impact on prognosis, recurrence and life expectancy [8,11,14,21,22]. Although the reported variability of PNI incidence might be influenced by the lack of standardisation of surgical techniques and pathological processing, the reported incidence of intrapancreatic PNI ranges from 76.2% to 97.8%, while the extra-pancreatic PNI varies from 52.2% to 75.8% [8]. Nerve-positive patients have a more elevated risk of death compared to patients without nerve infiltration [9]. It has been demonstrated that the nerve diameter affected by PNI affects prognosis. Specifically, the mean area of nerves in PDAC tissues is almost four times greater than in normal tissue [9]. In patients affected by PDAC, a nerve invasion greater than 8 mm is linked to a high frequency of positive resection margin [23]. Nerve infiltration is also a risk factor for patients without lymphatic metastases, indicating a subpopulation of N0 patients with an increased probability of death at an early age [9]. The density of the ganglia is more elevated in the pancreatic head as opposed to the body and pancreatic tail. The process of PNI in PDAC is yet to be completely elucidated, even though the participation of a great variety of molecules is known. Of these, neurothrophins, catecholamine, chemokines, matrix metalloproteinases and other mediators are released by tumour-associated macrophages, Schwann cells, pancreatic stellate and cancer cells [14]. Nerves play a clear role in pancreatic tumorigenesis [24]. There are three main mechanisms used by cancer cells to control nerves: axonogenesis (the enlargement of nerves), neurogenesis (growth of neural progenitors) and neural reprogramming (the transforming of a sensory nerve into an adrenergic nerve) [22]. Clinically, patients with hyperglycaemia show an increased PNI as opposed to individuals with euglycaemia affected by PDAC [14]. It is well known that obesity and diabetes can increase the probability of developing PDAC [14,25,26]. Hyperglycaemia may precede the diagnosis of PDAC for a mean period of 36-30 months [26]. Metformin improves the prognosis of PDAC by inhibiting the desmoplastic effects of activated pancreatic stellate cells (PSCs) and by phosphorylating 5′ AMP-activated protein kinase (AMPK) [25,26]. In PDAC, altered glycolysis has been recognized, and many glycolytic enzymes are associated with poor prognosis [26]. A high-fat diet promotes PDAC with molecular mechanisms that lead to chronic inflammation and fibrosis [26]. Modern observations document that PNI is more common in diabetic patients that show a lower frequency of abdominal pain [14]. Hyperglycaemia enhances NGF expression, neurotropism and cancer cell invasion in PDAC [11]. The involvement of nerves in PDAC is increasingly reported together with integrated molecular mechanisms. Although specific knowledge in the field of PNI has been reached, the translation into clinical practice is limited and complex. PNI has attracted a lot of interest and is to become a concrete potential medicinal target in the treatment of PDAC once comprehensive research in this field is completed.

## 3. Attractive Molecular and Cellular Signalling Pathways of PNI in PDAC

PNI, the phenomenon of pancreatic cancer cell invasion along nerve tissues, is based on the reciprocal interaction and interface between cancer cells and nerves. This paracrine cell signalling is based on the release of several molecules and the activation of specific perturbed pathways (Figure 2). PNI, together with the proliferation of fibrotic tissue under the stimuli of immune cells, PSCs and extracellular matrix, defines the desmoplastic cascade in PDAC. The mechanism of PNI in PDAC is not fully clear. Modern studies confirm how many neurotrophins are implicated in enhancing cancer–cell interaction and promoting PNI. PDAC is connected to the sympathetic nervous system (SNS) by a positive feedback loop. The SNS’ intrapancreatic neurons supply norepinephrine (NE) which modulates the β2-adrenergic receptors (β2-AR) of PDAC cells producing an upregulation of nerve growth factor (NGF). NE/β2-AR signalling promotes PNI by inducing epithelial–mesenchymal transition (EMT) and upregulating metalloproteases, MMP-2 and MMP-9, specifically [10]. In addition, NE promotes pancreatic PNI through β-AR/PKA/STAT3 signalling [10,11]. The nervous system releases serine into the pancreatic tumour microenvironment, but in the case of serine scarcity, more NGF is produced by PDAC cells by increasing the progression of axons along the tumour [10]. Axonogenesis is enhanced by the binding of NGF to tropomyosin receptor kinase (Trk) receptors. Finally, the augmented nerve density within the PDAC tumour increases NE production, so that a positive feedback loop is triggered, promoting PDAC innervation [27].

In brief, NGF, the Brain-Derived Neurotrophic Factor (BDNF), Neutrophin 3 (NTF3) and Neutrophin 4 (NTF4) are the major neurotransmitters secreted by neural and tumour cells in PDAC [10,11,14,28] (Table 1). NGF is also produced in PDAC cells and might be secreted by tumour-associated immune cells [10]. NTF3 is overexpressed in human PDAC, and with its receptor TrkC, is highly expressed in PDAC nerves [28]. Its high-affinity receptors belong to the Trk family, while the low-affinity receptor is the p75 neurotrophin receptor (p75NTR) [10,28]. NGF levels are upregulated in PDAC compared with their normal counterpart [13]. Moreover, the expression of NGF and its receptor TrkA correlates with the incidence of PNI, lymph node metastasis and negative outcome [14,28]. Survival is also influenced by different neurotrophin signals. A high expression of neurotrophic receptor tyrosine kinase 1 (NTRK1) is linked to poor prognosis, while an overexpression of NGFR correlates with a long survival [11]. In addition, glial cell line-derived neurotrophic factor (GDNF), neurturin (NRTN), artemin (ARTN) and persephin (PSPN) are neutrophins secreted by glial cell in neurons [10,14,21,28]. Nerves, Schwann cells, and macrophages secrete GDNF [13]. They bind to GFRα1, GFRα2, GFRα3 and GFRα4, respectively. Increased levels of ARTN and its receptor GFRα3/RET are documented in PDAC compared to the normal pancreas [14]. The ARTN–GFRα3 pathway promotes cell invasion [11]. Although the BDNF and its receptor TrkB are overexpressed in human PDAC, its levels did not correlate with increased PNI [21,28]. GDNF expression is frequently found in PDAC patients with PNI, and the GDNF–GFRα1–RET axis is responsible for PDAC metastasis [11]. Pleiotrophin (PTN) and its receptor SDC3 regulate neuroplasticity during the PNI cascade [11]. CXCR4 expression correlates with PNI in human PDAC [11]. The neural tropism of PDAC is decreased by blocking the CXCL12/CXCR4 pathway [10]. In an in vivo model, blocking the CXCL12/CXCR4 signal might reduce the tumour size and PNI development [11]. Interestingly, the CXCL1/CXCR1 pathway correlates with the occurrence of PNI [14]. In addition, neural chemokine fractalkine (CX3CL1) and the receptor CX3CR1 are involved in PNI [13]. PNI and the spreading of tumour cells along intrapancreatic and extra-pancreatic neurons are modulated by the CX3CR1/CX3CL1 axis [10].

CX3CR1 is present in PDAC cell lines such as MIA PaCa-2, CFPAC-1, PACA44, T3M4, PANC-1, AsPC-1, A8184 and in PDAC patients while CX3CL1 is expressed in nerves [11]. Tumour-associated macrophages (TAMs) may promote PNI and are found around nerves invaded by pancreatic cancer cells. Mucin-1 (MUC-1) and its receptor myelin-associated glycoprotein (MAG) are found in high levels in PDAC cells. Both of them participate in PNI by interacting with Schwann cells [13,14,21]. Mucin-4 (MUC-4) is overexpressed in PDAC [7]. MUC-4 is implicated in the cross-talk PDAC cells–nerves by regulating netrin-1 expression through the HERs/AKT/NF-κB pathway [7]. In this scenario, human PSCs also have a neurotrophic role in regulating PNI. By exploring the molecular mechanisms of PNI, a critical role is recognized in Pleiotrophin (PTN). PTN and its high-affinity receptor N-syndecan intensify PNI in PDAC [14]. Synuclein-γ expression is related to PNI and may upregulate the expression of MMPs [13,14,21]. Recent investigations have found that MMPs are important contributors to PNI [14]. The L1 cell adhesion molecule (L1CAM), also known as CD171, and neural cell adhesion molecule (NCAM), also called CD56, regulate neural adhesion/migration and are expressed by Schwann cells and pancreatic cancer cells [10] (Table 1). Their expression in PDAC patients is associated with neuropathic pain, nerve invasion and poor outcome [29]. Specifically, L1CAM upregulates MMP-2 and MMP-9 by activating STAT3 [10,29]. A role in the generation of PNI in PDAC patients is also demonstrated by kinesin family member 14 (KIF14) and Rho-GDP dissociation inhibitor beta (ARHGDIbeta) [13]. A leukaemia inhibitory factor (LIF) is involved in neural remodelling in PDAC. LIF and its receptors (GP130 and LIF receptor) are expressed in nerves [7].

## 4. PNI and Pain Generation in PDAC Patients

In clinical practice, patients with PDAC present abdominal or back pain [30]. Pain is the consequence of pancreatic enzyme insufficiency, obstruction of the pancreatic duct and PNI. Multiagent chemotherapy improves survival and decreases pain levels in PDAC patients. Gemcitabine improves the alleviation of painful symptoms compared with 5-fluorouracil (5-FU), but the benefits in preserving the quality of life are also documented with second-line chemotherapy [30]. Advanced radiation therapies, such as stereotactic body radiation therapy and conformal radiotherapy, alleviate painful symptoms in PDAC patients with mechanisms that interfere with the disruption of the inflammatory pathways and the formation of desmoplastic reaction [30]. Most PDAC pain is attributable to PNI signalling. In 1999, a strong correlation between NGF expression, the recurrence of PNI and the level of pain sensation was documented for the first time, and a possible responsibility of PNI in pain origination was recognized [13]. Additional studies confirmed the correlation between abdominal pain in PDAC patients with high levels of NGF and its receptor TrkA [10]. Around 80% of patients with PDAC report moderate to severe pain intensity levels [30]. Of these, more than 30% are hospitalized for pain management [30]. Many molecules in PNI play a role in pain signalling. NGFs produced by immune cells stimulate pain via binding with TrkA or P75 NTR [14]. Several neurotransmitters, such as glutamate, Substance P (SP), NGF and calcitonin gene-related peptide (CGRP), are implicated in pain generation in PDAC patients [30]. Nerves secrete CGRP and SP after the stimulation of the transient receptor potential cation channel subfamily (TRPV1) [31]. These molecular pathways contribute substantially to the origination of pain [7,31]. The overexpression of TRPV1 is closely linked with pancreatic pain [31]. SP mRNA expression levels are increased in patients with chronic pancreatitis. SP serum values are decreased with surgical resection as opposed to pre-operative levels. This means that after surgical resection of the inflammatory tissue mass, the levels of inflammatory mediators also decrease in the serum of patients, with consequent benefits [32]. Molecular observation suggests that SP and its receptor, NK-1R, play a part in the local inflammatory response in patients with inflammatory bowel diseases, particularly in cases of ileal Crohn’s disease [33]. While NGF and GDNF cascade stimulates neuropathic pain via enhancing TRPV1 levels, the opposite effect is observed with ARTN [10,14]. Interestingly, the downregulation of TRPV1 and P2Y receptors produced by a natural compound, Honokiol (HNK), extracted from the Magnolia plant, has been found to have a pain-relieving effect by downregulating TRPV1 and P2Y nociceptors [34,35]. These initial findings indicate that HNK could act as a novel inhibitor able to suppress PNI in PDAC. Cancer-associated pain might be reduced by the neutralisation of chemokine CCL2, as reported in a clinical trial [36]. After measuring the levels of chemokines, authors have discovered that the chemokines CCL21 and CXCL10 are able to stimulate pancreatic cancer cells’ migration towards nerves. In in vivo models and in patients with PDAC, the hypersensitivity can be decreased by inhibiting these proteins [36]. L1-CAM could play a neuropathic pain generation role by activating p38 MAPK pathways [10].

## 5. Role of Extracellular Vesicles in PNI

Extracellular vesicles (EVs) are small membranous structures released from cells that comprise exosomes, microvesicles and apoptotic bodies [37]. In the past few decades, they have been the object of intense studying because of their participation in key physiological roles, as well as being involved in the pathogenesis of a plethora of diseases, including cancer [38]. EVs can be divided in different groups according to their size and biogenesis [39]. The apoptotic bodies are larger in size, about 500–2000 nm in diameter [39], which derive from cells undergoing apoptosis and participate in the immune function [40]. EVs that are 100–1000 nm in size are called microvescicles and were discovered in the 1980s; they develop from an outward budding of the plasma membrane, which is reflected by their contents [39]. Discharged by almost all cell types, they are located in body fluids such as blood, urine, saliva, cerebrospinal fluid, amniotic fluid and breast milk, and have been discovered to carry out many diverse functions [41,42]. Their lipid bilayer membrane includes proteins, lipids and nucleic acids [38]. The smallest EVs (30–150 nm of diameter) are called exosomes or small EVs [38]. Their origin is different from that of microvesicles and apoptotic bodies, as they are borne by means of an endocytic pathway [40]. Exosomes’ content reflects that of the cell of origin, even though some molecules, particularly those involved in signal transduction and cell metabolism, can be common between exosomes derived from different cell types [43]. Cell signalling and cell–cell communication throughout the body are the main physiological roles carried out by exosomes. They can contribute to the immune system’s activation by stimulating macrophages to make pro-inflammatory cytokines, such as tumour necrosis factor (TNF) and stimulating natural killer cells and dendritic cells [44,45,46,47], and are also implicated in pathological conditions, such as neurodegenerative diseases [48]. Their contribution to the spread of cancer in many different ways has also been studied. Cancer cell exosomes contribute to tumour microenvironmentmodulation and to the development of pre-metastatic niches [49]. In addition, they can also influence metastasis formation, boosting cancer progression. To speed up tumour migration and invasion, tetraspanins carried by exosomes can modulate the extracellular matrix, specifically with proteases and integrins [50]. Tumour microenvironment immunoresistance in pancreatic cancer is also attributed to exosomes’ signalling [51]. This and chemoresistance, although still poorly understood, could be due to exosome-carried molecules affecting pancreatic cancer cells’ response to drugs [40]. In addition, angiogenesis of pancreatic cancer could be also attributed to the role of exosomes [40]. Molecules transported by exosomes are currently being studied as new diagnostic biomarkers for PDAC [52] and, due to their small size, structure and immunological tolerance, as a drug delivery system [53]. Among all the roles EVs have been discovered to be involved in, neurological processes, both physiological and pathological, have also been found to be modulated via the intervention of these small structures. Cancer–nerve cross-talk and tumour PNI have recently been looked at as events which can been explained by an involvement of tumour-derived EVs (Figure 3).

Recent studies on the role of exosomes in cancer propagation have indicated that these small vesicles released from cancer cells can stimulate tumour innervation [22]. Enhanced axonogenesis has been observed in PC12 cells when those were treated with exosomes derived from cancer cells, as opposed to exposure to normal cell-extracted exosomes, and treated with pharmacological exosome inhibition and the blocking of genes responsible for exosomes release [54]. This increased neurogenesis has been attributed to the unique exosomal molecular and, specifically, dysfunctional miRNA cargo [22,55]. In a study using PC12 cells, mice and patients’ material, the authors established that EphrinB1, a molecule carried by cancer-derived exosomes, was responsible for potentiating nerve growth in tumours [56]. EV extracted from the fallopian tube cell line FT33-tag modified to express Myc and Ras oncogenes were found to stimulate neurogenesis in PC12 cells, suggesting the importance of specific EV cargo in this phenomenon [57,58]. In addition, observations of EV derived from cell lines treated with cisplatin revealed that exposure to chemotherapy induces cells to produce EVs with altered cargo responsible for increased neurogenesis compared to exosomes collected from cells not treated with cisplatin [56,58]. In a study on colorectal cancer, EVs extracted from the serum of patients with or without PNI and healthy subjects were compared. EVs from the two group of patients contained different proteins, and the EV-carried protein stratifin was identified as a diagnostic and prognostic indicator of PNI in this type of tumour [59]. The growth factor midkine has been shown to promote PNI in pancreatic cancer [60]. Midkine was proposed as a biomarker of PNI, associated with tumour stage and diabetes [61]. Interestingly, we recently found that EVs derived from pancreatic cancer cells are enriched in midkine [37]. Some authors have recently demonstrated how disrupting the communication between cancer cells and nerves reduces nerve infiltration in tumour tissues, slowing down cancer progression. Using nanoparticles containing bupivacaine, a non-opioid analgesic, the researchers were able to block cytokines and neurotrophic factors produced by breast cancer cells, reducing neuron cell growth and obtaining a decrease in tumour size in a mice model of breast cancer [62]. Therefore, given the well-known role of EVs in cell-to-cell communication, targeting cancer-derived EV to block PNI could be an interesting therapeutic approach to counteract cancer.

## 6. Conventional and Experimental Treatments for PNI

Patients treated with neoadjuvant therapy and pancreatectomy have a different prognosis according to the presence or absence of PNI, which is an important factor in the progression of PDAC [23]. PNI correlates with pancreatic tumour size, resection margins, post-therapy tumour stage, and lymph node metastasis. In subjects who received pancreatectomy, the reported frequency of PNI ranges from 70.8 to 93.0%, while in PDAC patients treated with neoadjuvant therapy and pancreatectomy PNI is present in 43–58% of cases [11,14,21,23]. This is unequivocal proof that neoadjuvant therapy significantly decreases PNI. Specifically, the neoadjuvant scheme based on FOLFIRINOX treatment shows a decrease in PNI and the number of positive lymph nodes [8].

### 6.1. Surgical Treatment for PNI

The aim of pancreatic surgery is to achieve a curative resection (R0); however, PNI is also detected in the majority of cases after R0 surgery [63]. Unfortunately, the most common site of residual cancer (R1) is the retroperitoneum and the mesopancreas [64]. Surgery has limitations in its curative intent when pancreatic cancer cells have invaded nerves and the retroperitoneal layer, as documented in advanced tumour disease. Unfortunately, in advanced PDAC, the infiltration of tumour cells on neuro-vascular structures is very common and correlates with a worse prognosis [63]. All the validated surgical procedures have a curative intent, and this means that microscopic pancreatic resection margins have to be negative for cancer cell infiltration [65,66]. The peripancreatic nerve plexus needs to be resected. According to this concept, surgical treatment has progressively evolved. The traditional Whipple operation focused only on the excision of a tumour was modified with additional surgical techniques that have improved the extension of lymph node dissection and complete retroperitoneal resection with or without portal vein/mesenteric vascular resection associated with their reconstruction [65]. The importance of including the surrounding connective tissue with the extra-pancreatic nerve plexus is well known to clinicians. Therefore, the surgical procedure has evolved with the introduction of the “en bloc resection with no touch tumour principle”. The extension of resection has included the total mesopancreas structure [64]. Current efforts are moving in the direction of improving intraoperative PNI detection with intraoperative ultrasound/intraductal ultrasound and with optogenetic neuron-staining techniques, together with fluorescence, which are useful in identifying neural structures [64]. However, these data are insufficient to design clear conclusions that can be translated into clinical practice. The denervation of the pancreas by using an ethanol-induced celiac plexus block is a useful strategy to prolong the survival of patients affected by advanced PDAC [67]. Patients that received an ethanol-induced splanchnicectomy, a mechanical procedure that consists in the surgical resection of splanchnic nerves, survived longer than the control group (median 9.15 vs. 6.75 months) [67]. According to these results, the surgical or chemical ablation of pancreatic innervation reduces painful syndromes and improves quality of life. Pancreatic denervation is an accepted therapy for pain control both in patients with PDAC and in those affected by chronic pancreatitis. Unfortunately, although the effects on the neurolytic celiac plexus are useful in pain control, they do not affect survival [24]. Further studies are required to confirm the controversy of retroperitoneal nerve dissection in PDAC and the clinical role of technology-assisted clearance. Modern data documented that adjuvant chemotherapy improves the outcome of patients with PNI after R0 resection, whereas patients without PNI did not benefit from adjuvant treatments [63]. Consequently, most of the treatment methods described in this review are included in targeted drug therapy. Surgical resection has a palliative role in advanced PDAC. In the future, the timing of upfront surgery or systemic therapies must be carefully revised according to the evidence that PNI is a common and early feature also in localized pancreatic tumours. In addition, more efforts must be made in mapping nerve structure and in improving pancreatic tumour classification, with particular emphasis on the redefinition of curative resection. These data confirm how the dynamics of PNI have to be fully investigated to better standardize patients’ risk and multimodal treatment strategies.

### 6.2. PNI-Targeted Chemotherapy

Advancements in PNI-targeted therapy have been achieved, and several new drugs have been investigated. NGF inhibition by using specific antibodies or gene silencing reduces cancer progression, metastatic development and pain in a pre-clinical model of PDAC [28]. The NGF–TrkA signalling pathway has great potential as a therapeutic target, and the treatment approach consists in using antibodies [10,14]. By blocking the tyrosine kinase receptor (TRK) using pharmacological inhibitors, it is possible to relent tumour advancement (Figure 4) [68].

Anti-NGF therapy is able to suppress TRPV1, Substance P and CGRP in a mouse model of PDAC, increasing pain relief [7]. The usual treatment approach consists of impeding the binding of NGF to its receptor protein TrkA using antibodies such as muMab911, Tanezumab, MNAC13, PHA-848125 and ARRY-470 [14] (Table 2). PHA-848125 is an inhibitor of TrkA and cyclin-dependent kinases (CDKs) that is able to reduce PDAC development. In addition, PHA-848125 reduces the painful syndrome in PDAC patients and represents a prospective target for PNI treatment [11,14,69]. Phase I and II clinical trials involving PHA-848125 are in progress [69]. The inhibition of TrkA, B and C by the AZD1332 agent in combination with radiotherapy has shown encouraging results on pancreatic cancer growth in vitro, but these data were not confirmed in in vivo xenograft models [10]. Tanezumab, a monoclonal antibody against NGF, has already been tested in a phase III trial for analysing the effects on pain intensity in bone metastatic patients (NCT02609828) [22]. In orthotropic PDAC models, a Trk–NGF inhibitor (LOXO-101) is able to decrease innervation and slow down PDAC [70]. Treatment with LOXO-101 in mice on a gluten-free diet achieved a decrease in tumour weight, spreading and neuronal innervation [70]. Therefore, inhibiting Trk blocks tumour innervation impeding the necessary metabolic contribution that PDAC cells need in an environment lacking Serine/Glycine.

Based on pre-clinical studies, CEP-701, an antagonist of tyrosine kinases including Flt-3, TrkA/B and JAK-2, was tested together with gemcitabine in subjects with late-stage PDAC in a phase I trial. Although the combination was tolerated, the benefits of CEP-701 were limited and the trials were not further pursued because of insufficient efficacy [71]. Entrectinib (RXDX-101) is a selective antagonist of the tyrosine kinases TrkA, TrkB, TrkC, ROS1 and ALK encoded by NTRK1, NTRK2, NTRK3, ROS1 and ALK, respectively. The oral administration of RXDX-101 showed a reduction of pancreatic tumours or stable disease [72]. VMD-928 is a novel allosteric and irreversible TrkA selective antagonist presently studied in a human phase I trial. This study includes a cohort of patients with PDAC [73]. Another Trk inhibitor, TSR-011, also active against ALK, has been found to be effective on tumours resistant to ALK inhibition. The results are under investigation in a Phase I clinical trial [74].

AZD1332 is a selective small-molecule antagonist of the TRKtyrosine kinase family displaying a strong ATP-competitive blocking of the three TRK receptors. It was studied for the first time in PDAC tissues after the effects of combined radiation therapy [76] (Table 2). Among neurotrophins, slit-guidance ligand 2 (SLIT2) is down-regulated both in cell lines and in patient tumour tissue [11]. In other terms, stopping the interplay between SLIT2 and its receptor ROBO1 enhances the invasion of PDAC cells confirming the natural inhibitory role of SLIT2. Resiniferatoxin (RTX), an analogue of capsaicin, has been found to promote apoptosis in pancreatic cancer cells and to downregulate pain awareness [85]. The effect of Resiniferatoxin, a TRPV1 agonist, is analysed in in vitro models by using MIA PaCa-2 and Capan-1 cancer cells. These data confirm that Resiniferatoxin promoted apoptosis in pancreatic cancer cells and might represent a new and efficacious strategy for PDAC patients against neurogenic pain [85]. TRPV1-targeted drugs have opened new insights into PNI therapy. The inhibition of the NGF receptor has been shown to potentiate the effect of gemcitabine, a chemotherapeutic agent commonly used in clinical practice [9]. Injection of NGF or Immunoglobulin G influences the progression and metastasis of PDAC [67]. After the use of an anti-NGF inhibitor, the indicators of neurogenic inflammation, such as SP and CGRP, were significantly reduced in a genetically engineered model of PDAC [67]. Anti-NGF therapy does not directly affect PDAC disease; however, it reduced tumour cell mobility and metastases in in vivo studies when mice started anti-NGF treatment at 8 weeks of age. Interestingly, anti-NFG treatment in KPC mice decreases gene expression involved in nociception and cancer cell invasion, such as TrkA, NGF receptor, tachykinin precursor-1 and calbindin-1 [7]. Anti-NGF treatments could be a potential neoadjuvant therapy used with the aim of restricting pancreatic tumour cells [67]. Anti-NGF siRNA encapsulated in nanoparticles has the ability to decrease PDAC growth in a mouse model [9]. L1CAM is a mediator of PNI. In vivo studies revealed that the anti-L1CAM Ab treatment significantly reduces PNI in a KPC transgenic animal model [11,29]. The inhibition of L1CAM decreases the nerve density and the ability of cancer cells to invade nerves by the deregulation of MAPK pathway activity with effects depending on STAT3 phosphorylation [29].

Non-selective β-blocker therapy reduces primary tumour growth, pancreatic cancer cell dissociation and increases survival [7]. In a preclinical study (NCT00502684), β-blockers, such as propranolol/propranolol hydrochloride, reduced PDAC-specific mortality [10]. In a multivariable analysis, treatment with beta-blockers did not improve survival in PDAC patients [77]. Using an in vivo model of PNI, it was possible to show that cancer cells’ neural spreading might be treated by the administration of an attenuated, replication-competent, oncolytic herpes simplex virus. The utility of the injection of the attenuated virus is in the intraoperative detection of invaded nerves, by using fluorescent imaging, which finally might be resected [13]. Oncolytic viruses are a novel antitumour therapy against PDAC [11]. Recently, it has been reported that oncolytic adenoviruses (OBP-301 and OBP-702) repressed the migration and invasion of pancreatic cancer cells via the induction of p53 expression, autophagy and apoptosis [86]. Specifically, OBP-702 acts on ERL signalling, producing a blockage of PDAC cells’ migration and invasion caused by a neurosecretory factor [86]. Further research is needed to confirm the safety of oncolytic viruses, such as OBP-702, in PDAC patients. In athymic need mice, the treatment with Botulinum toxin (Botox) significantly decreases the pancreatic tumour size and the apoptotic rate. Honokiol (HNK) is a polyphenolic compound extracted from Magnolia species [34]. HNK attenuates PNI by inhibiting the invasion, migration and EMT (epithelial to mesenchymal transformation) of pancreatic cancer cells. HNK can inhibit PNI in PDAC by suppressing the activation of SMAD2/3 and the cross-talk between nerves and cancer cells. Specifically, the knockdown of SMAD2 or SMAD3 can lead to the inhibitions of NGF and BDNF in pancreatic cancer cells [34]. Several CXCR4 antagonists are presently under investigation in phase I/II trials [82]. The CXCL12 peptide analogue CTCE-9908, an inhibitor of CXCR4/CXCL12, was tested in patients with solid cancers [82] (Table 2). Unfortunately, VEGF inhibitors, radiotherapy or taxanes might upregulate CXCR4/CXCL12 expression in several tumours, enhancing the invasiveness and the metastatic behaviour. MiRNAs, defined as single-stranded non-coding RNAs, might promote PNI [15]. In PNI of PDAC, miR-429, miR-133a and miR216 are downregulated while miR-191, miR-21, miR-23a/mir-27a and miR-17 are upregulated [15]. The inhibition of miR-21 might reduce PDAC growth in in vitro and in vivo models together with the inhibition of miR-133a by LncRNA XIST that, by upregulating the EGFR expression, might promote PNI in PDAC [15]. Specifically, miR-23a and miR-27a might promote PNI in PDAC, and their synergistic inhibition might represent a key therapeutic point in future research. This promising research might offer future valid cancer therapeutics.

## 7. Cannabinoids in Pancreatic Cancer Treatment

Cannabinoids have recently attracted attention for their potential as anticancer agents for many types of cancers, including pancreatic cancer [87,88]. However, the anticancer activity of medicinal cannabinoids is still controversial, and the efficacy in clinical setting is unproven [88]. Interestingly, medicinal cannabis and cannabinoids have been proposed for the treatment and management of cancer pain [89]. Cannabinoids found in the human body, or endocannabinoids, regulate many physiological functions such as learning, memory, pain and appetite. These effects were thought to be mediated by the endocannabinoid system (ECS), mainly composed of two G-protein-coupled receptors (GPCRs), namely cannabinoid receptors 1 and 2 (CB1 and CB2). These were expressed in different cell types and activated by two endogenous lipids, endocannabinoids anandamide (AEA) and 2-arachidonoyilglycerol (2-AG), alongside several catalysts and degradative enzymes [87]. Increasing evidence demonstrated that the response to cannabinoids is mediated not only by the canonical receptors CB1 and CB2 but also by other G-protein coupled receptors, such as GPR55 and GPR119, transient receptor potential vanilloid (TRPV) channels, such as TRPV1, and peroxisome proliferator-activated receptors (PPARs). These receptors, alongside emerging bioactive lipids and enzymes regulating their synthesis and degradation, are collectively known as the “expanded cannabinoid system” or “endocannabinoidome” [87,90]. The endocannabinoidome is a complex system whose knowledge is in constant evolution, developed particularly in the central nervous system but also spread in the periphery of the body and in the gut, where it participates in the regulation of the enteroendocrine system’s functions [90].

Many receptors belonging to the endocannabinoidome family, CB1, CB2, GPR55, GPR119, TPRV1 and PPARs, have been found to be overexpressed in pancreatic cancer cells/tissues compared to healthy controls and/or favour tumour growth [85,91,92,93,94]. CB2 receptors showed antitumour effects in pancreatic cancer cells via an endoplasmic reticulum stress-regulated protein p8 dependent mechanism and activating transcription factor. In addition, cannabinoids are involved in the regulation of the immune system in cancer. Cannabinoid receptors are involved in decreasing the activation and migration of pancreatic stromal cells by pancreatic cancer cells in chronic pancreatitis [87]. Selective CB1 and CB2 receptor agonists potentiate the anticancer action of gemcitabine by increasing the ROS-mediated growth inhibition. The synthetic cannabinoid agonist WIN55212,2 decreases pancreatic stromal cells’ migration that participate in cancer invasion, immunomodulation and chemoresistance. Although studies on cannabinoid targets in PDAC are limited and this field of investigation should be fully explored, evidence points to a potential for strategies directed at exploiting the endocannabinoidome as an antineoplastic tool; understanding the role of cannabinoids could boost innovation in pancreatic cancer therapy.

When used in combination with chemotherapy, cannabidiol (CBD), one of the major compounds found in the cannabis plant, has been found to increase its efficacy in vivo in a transgenic model of pancreatic cancer [92]. Indeed, CBD, in association with gemcitabine, enhances the anticancer efficacy in the KPC mouse model of PDAC. CBD potentiates gemcitabine’s action by inhibiting GPR55 receptor. Unfortunately, CBD/gemcitabine adjuvant therapy has not been sufficiently investigated in clinical practices so far. Interestingly, CBD enhances the radiotherapy efficacy in in vitro pancreatic cancer models [95].

Abdominal and back pain is a very common and unbearable symptom for pancreatic cancer patients that current therapies fail to manage satisfactorily, heavily affecting patients’ quality of life. Some receptors belonging to the endocannabinoidome, particularly TRPV1, located in pancreatic cancer cells and tissues, CB1 and CB2, GPR55, GPR199 and PPAR-α, have been found to be involved in pain regulation [96,97,98,99]. Therefore, using cannabinoids to target these receptors could perform a dual function by not only counteracting cancer progression, but also by being a valid course of action to alleviate the burden of pain in pancreatic cancer. To this end, some progress has already been made, but the potential exploitation of the endocannabinoidome system for the treatment of pancreatic cancer pain remains largely unexplored. CB1 subtypes are involved in processing nociceptive signals while CB2 can decrease inflammation and produce endogenous opioids. Olorinab is a CB2 agonist potentially useful in the treatment of visceral gastrointestinal pain that currently is in a phase II clinical trial (NCT04043455) [100]. A THC- and CBD-based drug, Nabiximol (CBD:THC 1:1), is already used to treat neuropathic pain in multiple sclerosis [101], and a decrease in neuropathic pain has been achieved administering this medication to cancer patients in a phase III trial (https://clinicaltrials.gov/ct2/show/results/NCT01361607; accessed on 23 November 2022). Nabilone and Dronabinol, synthetic cannabinoids, are used in palliative care to control suffering [102]. A recent study on neuropathic pain in rats reported the role of GPR55 and GPR199 in controlling this type of hyperalgesia [103]. In addition, the inhibition of GPR55 decreased pain caused by inflammation and chronic neuropathic pain in two different studies conducted on rats [104,105], and GPR55 antagonist CID16020046 injected into rats’ brains succeeded in reducing pain caused by inflammation, demonstrating GPR55 involvement in the nociceptive signalling [106].

Numerous studies have confirmed the involvement of TRPVs in the mechanism of nociception. TRPV1 and TRPA1 have been found to be involved in causing pain in diabetic rats affected by peripheral neuropathy [107]. In an animal study, the activation of TRPV1 in the spinal cord by capsaicin produced excessive glutamate liberation causing acute pain [108]. A randomized control trial on patients affected by neuropathic pain reported a reduction of pain after treatment with capsaicin patches [109]. Mice treated with the endocannabinoid-like molecule palmitoylethanolamide (PEA) showed a decrease in pain derived from peripheral neuropathy, while using synthetic antagonists to block PPAR-α and CB1 receptors reversed this effect, demonstrating the mediation of these receptors in this process [110]. A similar study experimenting with the efficacy of Camelina sativa on a rat model of visceral pain and other studies testing different compounds to alleviate inflammation and pain in IBD and rheumatic pain underlined the involvement of PPAR-α in the regulation of this types of pain [111,112]. These data corroborate the association between receptors belonging to the endocannabinoidome system and the mechanisms of the induction of pain in several different conditions, confirming the potential of manipulating the endocannabinoidome with the aim of managing neural pain in pancreatic cancer.

## 8. Conclusions

PNI is an omnipresent and ominous characteristic of PDAC that can be evaluated only after surgery. It is necessary to develop predictive biomarkers for monitoring PNI severity and make efforts to increase the pre-operative assessments of PNI. This might be helpful for surgeons that have to perform a curative resection and for planning a multimodal strategy. Our understanding of the nerve–cancer cross-talk in tumour progression is still limited since there is a paucity of valid PNI experimental models [113]. This is directly linked to the complex nature of the tumour–nerve interaction that involves the surrounding tumour microenvironment and probably the participation of different stromal cells. The role of nerves in the tumour microenvironment is often overlooked. In particular, further studies are required to reveal the emerging role of the neuro-immune axis in cancer progression. In addition, a better comprehension of the role played by exosomes in the nerve–cancer cross-talk could not only improve our understanding of PNI biology but also provide novel therapeutic opportunities by targeting specific molecules carried by exosomes. Future studies are advocated to better analyse the relationship between PNI and chemoresistance. Experimental drugs have the potential role of sensitizing tumour cells to conventional chemotherapeutics. The innovation of pancreatic cancer therapy might derive from understanding the role of innovative drugs and their value in combination with common chemotherapeutics to increase pain relief and to improve prognosis. PDAC is associated with excruciating pain that severely affects the quality of life of patients. Therefore, a better comprehension of the mechanisms and molecules involved in pain is crucial. In this context, the exploration of the role played by the endocannabinoids system is especially appealing. Indeed, specific cannabinoids may have the potential to target pain and tumour progression at the same time. However, some caution is necessary when targeting some molecules, such as neurotransmitters, as these are essential for many human physiological functions, and their neutralisation with specific antibodies may cause serious side effects. The future line of oncological research must be drawn with the conscience that the pancreatic tumour microenvironment is complex because different cell types with different roles compose it.

## Figures and Tables

**Figure 1 cancers-14-05793-f001:**
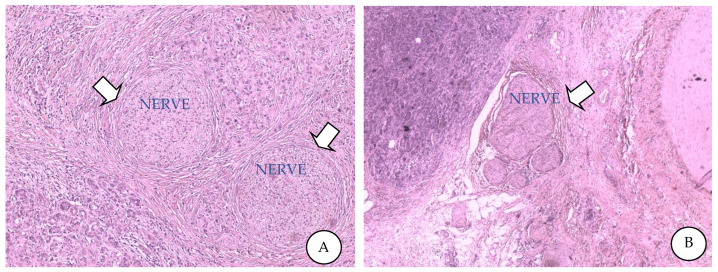
Perineural invasion from high-grade PDAC with desmoplastic and lymphocytic reaction into the surrounding stromal tissue (**A**). Hypertrophic nervous fibre bundle in the parenchyma away from PDAC cells (**B**).

**Figure 2 cancers-14-05793-f002:**
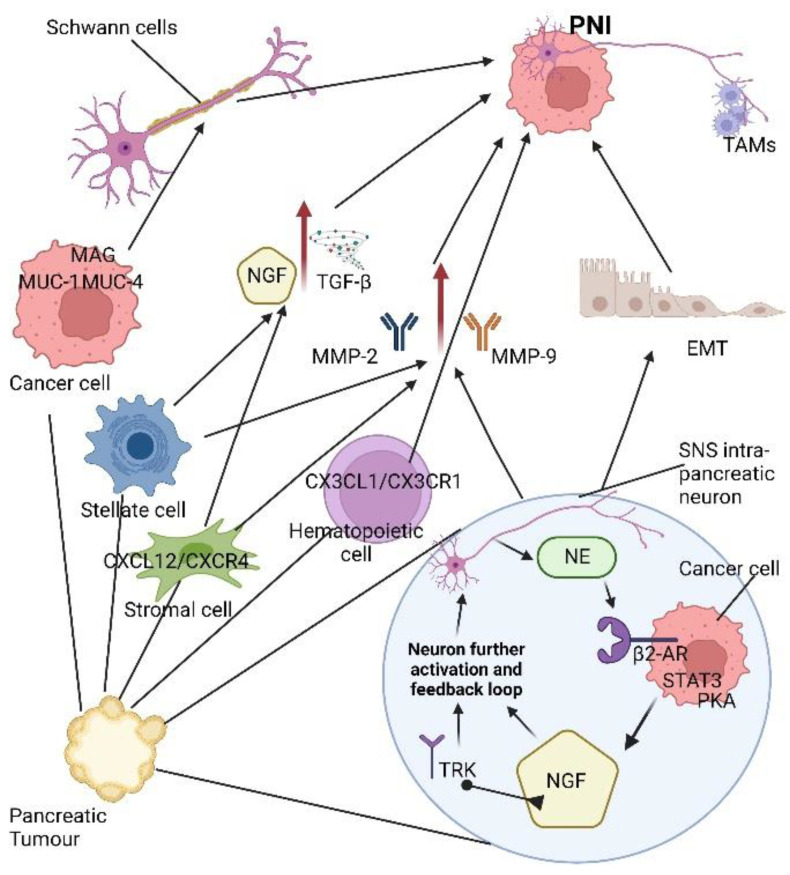
Perineural invasion signalling pathways in pancreatic cancer. β2-adrinergic receptor (β2-AR), epithelial- mesenchymal transition (EMT), chemokine (C-X-C motif) ligand 4 (CXCL4), C-X-C motif chemokine 12 (*CXCL12*), neural chemokine fractalkine (CX3CL1), neural chemokine fractalkine receptor (CX3CR1), matrix metalloproteinases (MMP-2, MMP-9), myelin-associated glycoprotein (MAG), nerve growth factor (NGF), norepinephrine (NE), protein kinase A (*PKA*), signal transducer and activator of transcription 3 (Stat3), sympathetic nervous system (SNS) transforming growth factor beta (TFG-β),transmembrane mucins (MUC-1 MUC-4), tyrosine kinases receptor (TRK), tumour associated macrophages (TAMs).

**Figure 3 cancers-14-05793-f003:**
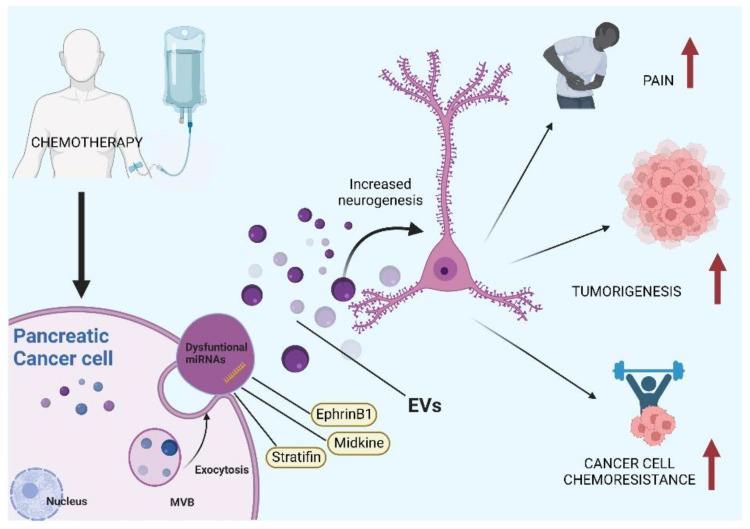
Possible role of extracellular vesicles in pancreatic cancer’s perineural invasion. Extracellular vesicles (EVs); multivesicular body (MVB).

**Figure 4 cancers-14-05793-f004:**
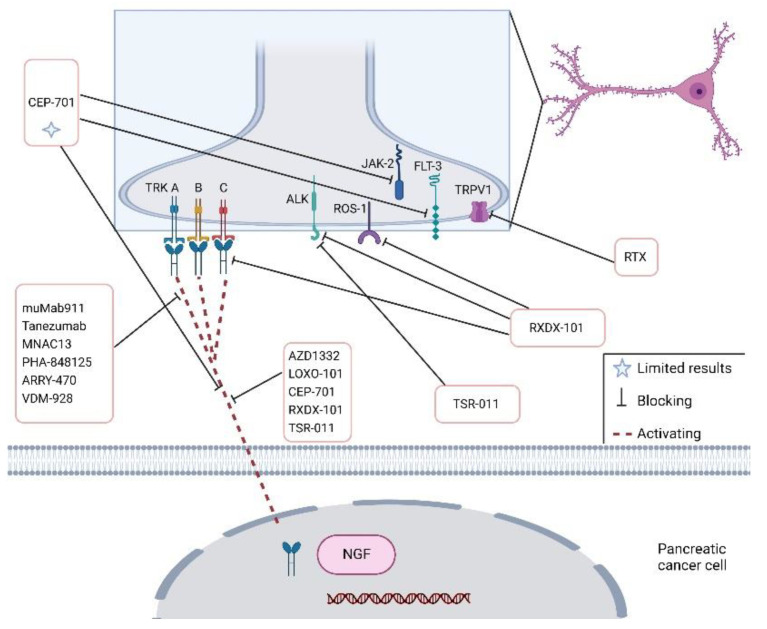
Schematic figure showing therapeutic agents and targets in pancreatic cancer’s perineural invasion. Anaplastic lymphoma kinase (ALK), FMS-like tyrosine kinase 3 (FLT-3), nerve growth factor (NGF), Janus kinase 2 (JAK2), proto-oncogene tyrosine–protein kinase-1 (ROS-1), Resininferatoxin (RTX), tyrosine kinases (TRK), transient receptor potential cation channel subfamily V member 1 (TRPV1).

**Table 1 cancers-14-05793-t001:** Molecular mechanisms of perineural invasion in PDAC.

*Molecule*	Receptor	Releasing Cell	Function	Inhibitor	Target	Effects
**NGF** Nerve Growth Factor	**NGFR** Nerve Growth Factor Receptor	Neuronal cells PDAC cells	cell growth cell survival cell maintenance-neurotrophic pain	Anti-NGF molecules	NGF SERINE	Inhibition of: -tumour growth -neurogenic inflammation -PNI
	**TRPV1** Transient Receptor Potential Cation Channels subfamily V member 1					
**BDNF** Brain-derived Neurotrophic Factor	---	Neuronal cells PDAC cells	-cell growth -cell survival -cell maintenance	Anti-NGF molecules	NGF SERINE	Inhibition of: -tumour growth -neurogenic inflammation -PNI
**NEUROTROPHIN 3**	**NTRK1** Neurotrophic Receptor Tyrosine Kinase 1	Neuronal cells PDAC cells	-cell growth -cell survival -cell maintenance	Anti-NGF molecules	NGF SERINE	Inhibition of: -tumour growth -neurogenic inflammation -PNI
**NEUROTROPHIN 4**	**NTRK2** Neurotrophic receptor Tyrosine Kinase 2	Neuronal cells PDAC cells	-cell growth -cell survival -cell maintenance	Anti-NGF molecules	NGF SERINE	Inhibition of: -tumour growth -neurogenic inflammation -PNI
**SLIT 2** SLIT–Guidance ligand 2	**ROBO1** Roundabout Guidance Receptor 1	PDAC cells CAFs (Cancer-associated Fibroblasts) (from PDAC cells)	-promotion of: cell navigation, Schwann cell migration (by Cadherine 2 pathway), neurite outgrowth -suppression of: cell migration, cell invasion	Anti-SLIT2-ROBO1	SLIT2/ ROBO1 signalling	-Motility and invasiveness of PDACs increase -Neural remodelling inhibition -PNI inhibition
**SERINE** (stimulated by NGF)	---	PDAC cells Neurons (Axons and DRG, Dorsal Root Ganglia)	-energy support	Anti-NGF molecules	NGF SERINE	---
			-PNI formation			
**GDNF** Glial-cell Derived Neurotrophic Factor	**RET 9** Proto-oncogene **RET 51** Proto-oncogene (expressed in PDAC cells)	Peripheral and central nervous system: Neural cells (Schwann cells and motor neurons) Macrophage	-KRAS signalling activation -Tumour growth maintenance -Migration of tumour cells to neural cells promotion -upregulation of MMPs - Neural invasion and metastasis promotion	KRAS-inhibitors PI3K-inhibitors	KRAS pathway PI3K	To inhibit tumour cell migration to neuronal cells
**PERSEPHIN**	**GFRα1** (RET co-receptor) GDNF family receptor alpha 1	Peripheral and central nervous system: Neural cells (Schwann cells and motor neurons) Macrophage	-KRAS signalling activation -Tumour growth maintenance -Migration of tumour cells to neural cells promotion -upregulation of MMPs - Neural invasion and metastasis promotion	GFRα1-inhibitors	GDNF– GFRα1– RET axis	To limit cell migration and tumour metastasis
**ARTN** Artemin	**GFRα3** GDNF family receptor alpha 3	Peripheral and central Nervous system: Neurons	-To trigger GFRα3-dependent invasion in PDAC cells -To drive tumour metastasis	GFRα3-inhibitors	ARTN– GFRα3 axis	To limit cell invasion and tumour metastasis
**Midkine**	**SDC3** SYNDECAN3 (on pancreatic nerves, neurons and Schwann cells)	PDAC cells	-Nerve proliferation and PNI Neuroplasticity regulation -Nerve damage after PTN accumulation (dual role of PTN–SDC3 in neuroplasticity during PNI)	Anti-Syndecan 3	PTN–Syndecan3 axis	PNI, nerveoutgrowthand proliferation inhibition
**PTN** **Pleiotrophin**	**SDC3** SYNDECAN3 (on pancreatic nerves, neurons and Schwann cells)	Necrotic PDAC cells	-Nerve proliferation and PNI Neuroplasticity regulation -Nerve damage after PTN accumulation (dual role of PTN–SDC3 in neuroplasticity during PNI)	Anti-Syndecan 3	PTN–Syndecan3 axis	PNI, nerve outgrowth and proliferation inhibition
**SEMA 3 D** Semaphorine 3D	**PLXND1** Plexin D1	Neurons	-Neuronal networks formation -Nerve density increase -Nerve invasion and PNI promotion	SEMA3D-inhibitors PLXND1-inhibitors	SEMA3D-PLXND1 axis	-Attenuation of the invasion of tumour cells towards the nerves -Nerve density decrease in tumour tissues
**CX3CL1** C-X3-C motif chemokine ligand 1	**CX3CR-1** C-X3-C motif chemokine receptor 1	Neurons and nerves	PI3K-AKT activation Chemoattractant for immune cells and neural cells Promoters of PNI process	CX3CR1-inhibitors	CX3CL1-CX3CR1 axis PI3K-AKT pathway	PDAC Inhibition PNI reduction
**CXCL12** C-X-C motif chemokine ligand 12	**CXCR-4** C-X-C motif chemokine receptor 4	Dorsal root ganglia (DRG)	Development and progress of PDAC Infiltration of immune cells in the tumour microenvironment	CXCR4-inhibitors	CXCL12-CXCR4 axis	Tumour size, nerve injury degree, PNI reduction
**CATECHOLAMINES** EPINEFRINE NOREPINEFRINE DOPAMINE	**ADRB2** Adrenoceptor beta 2 **PKA** Protein kinase CAMP-activated catalytic **STAT 3** Signal transducer and activator of transcription 3	Neural cells	Tumour invasion PNI promotion Regulator of pancreatic tumorigenesis Tumour stem cells proliferation Maintenance of an inflammatory tumour microenvironment	ADRB2-inhibition CAMP-activated catalytic-inhibitors STAT 3-inhibitors	ADRB2–PKA–STAT 3 signalling pathway	PDAC Reduction PNI inhibition
**IL-6 ST** Interleukin-6 signal transducer	**LIF** LIF interleukin-6 family cytokine	Schwann cells	---	---	---	---
**S P** Substance P	**KLRB1** Killer cell lectin-like receptor B1	CD8+ T-cells	PNI induction activating MAPK pathway	---	---	---
**LIF** LIF Interleukin-6 Family cytokine	---	Macrophage	---	---	---	---
**SNCG** Synuclein gamma	---	PDAC cells	To promote PNI and metastasis	---	---	---
**MUC 1** **Mucin 1** **(Cell surface-associated)**	---	**Pancreatic cancer cells**	---	---	---	---
**MAG** Myelin-associated glycoprotein	---	Schwann cells	---	---	---	---
**NCAM 1** Neural cell adhesion molecule 1	---	PDAC cells	To elicit structural changes in PNI cells, promoting PNI	Anti-NCAM antibodies	PDAC cells	To alleviate PNI
**L1CAM** L1 cell adhesion molecule	**STAT 3** Signal transducer and activator of transcription 3	Schwann cells	To enhance PNI-activating STAT3 pathway, promoting chemotaxis and upregulating the expression of MMP2 and MMP9	Anti-L1CAM antibodies	PDAC cells	To alleviate PNI
**CCL 2** C-C motif chemokine Ligand 2	**CCR 2** C-C motif chemokine receptor 2	Schwann cells Macrophages	Inflammatory macrophages recruitment from the circulation to the site of PNI	CCL 2-inhibitors	PDAC cells	To alleviate PNI

**Table 2 cancers-14-05793-t002:** Potential therapeutic targets in perineural invasion: conventional and experimental agents.

Target	Drug	Effects	Study	References
Trk-A/B	CEP-701	-	Phase I	[71]
TrkA, TrkB, TrkC	Entrectnib/RXDX-101	Stable disease/reduction	Phase I- II (NCT02097810); (NCT02568267); (NCT02650401);	[72]
TrkA, TrkB, TrkC	NOV1601(CHC2014)	-	Phase I NCT04014257	-
TrkA	VMD-928	-	Phase I (NCT03556228)	[73]
TrkA, TrkB, TrkC	TSR-011	-	Phase I-II NCT02048488.	[74]
Trk	DS-6051b	-	Phase I NCT02279433	[75]
TrkA, TrkB, TrkC	AZD1332	Increases radiosensitivity	Preclinical	[76]
βAR	β-blockers	No benefits on survival	Clinical	[77]
NGF	Tanezumab	-	Phase III NCT02609828	-
CXCR4	MSX-122 Inhibitor (partial antagonist of CXCR4)	-	Phase I NCT00591682	-
LIF	Ab-LIFR	Reduces PDAC-associated neural remodelling	In vitro (cocultures); In vivo (PDAC-bearing mice)	[78]
NGF	muMab911	Prevents hyperalgesia	In vivo	[79]
TrkA	MNAC13	analgesic effects	In vivo (CD1 mice)	[80]
TrkA	PHA-848125	Synergistic effects with Gemcitabine	Phase II	[69]
TrkA	ARRY-470	Reduces pain	In vivo (C3H/HeJ mice)	[81]
CXC4R/CXCL12	CTCE-9908	-	Phase I	[82]
TRPV1	Resiniferatoxin	Reduces pain	In vivo	[83]
Neuron ablation	Neonatal Capsaicin	Delays PanIN formation; prolongs survival	In vivo PKC mice	[67]
NGF	GNC–siRNA	Inhibits tumour progression	In vivo (subcutaneous model, orthotopic model and patient-derived xenograft model)	[84]

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
