# Peer review of "Perineural Invasion in Pancreatic Ductal Adenocarcinoma: From Molecules towards Drugs of Clinical Relevance"

_cancers, 2022, doi:10.3390/cancers14235793_

Round 1

Reviewer 1 Report

Overall, this review mainly summarized the molecular mechanisms and potential therapeutic targets of perineural invasion (PNI) in pancreatic ductal adenocarcinoma (PDAC). It provides a useful collection for the specialists in the field and shed the light on future research. I would be glad to recommend its publication on Cancers after the following of my comments are appropriately addressed.

Major concern:

1. I really appreciate the two tables summarizing molecular mechanisms and potential therapeutic targets of PNI in PDAC. However, no schematic diagram was shown in this review. I highly recommended the authors to include schematic figures showing PNI related molecular mediators discussed in the manuscript and their functional implications in their associated pathways in PDAC. Besides, it’s better to label therapeutic targets with conventional and experimental agents on the schematic figure. The overlapping target proteins, signaling pathways, and functions between different molecular mediators without agents can be potential targets for further study, and this message should be delivered to the readers. The authors can refer to Li J, Kang R, Tang D. Cellular and molecular mechanisms of perineural invasion of pancreatic ductal adenocarcinoma[J]. Cancer Communications, 2021, 41(8): 642-660.

2. Most of the main text are simple descriptions, lacking insight or summary. A good review should also be critical, summarize important research under the topic, identify consistencies and more importantly inconsistencies, and provide open questions for future research. For example, in the authors’ opinion, how should the field further explore the therapeutic targets of PNI in PDAC using single cell sequencing methods and other novel technologies?

3. PNI is a common feature of different cancer types. Could the authors summarize PDAC-specific as well as broad-spectrum molecular mechanisms and potential therapeutic targets of PNI.

Minor concerns:

1. The authors should spell out PNI (perineural invasion) when first mentioned in the abstract.

2. The authors should perform a thorough formatted and grammatical review for the manuscript and correct all typos. Eg, Line 105: “52,2% to 75,8%” should be “52.2% to 75.8%”

Author Response

Response to Reviewer 1

Comments and Suggestions for Authors

Overall, this review mainly summarized the molecular mechanisms and potential therapeutic targets of perineural invasion (PNI) in pancreatic ductal adenocarcinoma (PDAC). It provides a useful collection for the specialists in the field and shed the light on future research. I would be glad to recommend its publication on Cancers after the following of my comments are appropriately addressed.

We would like to take this opportunity to thank the Reviewer for his/her comments that have helped to improve our manuscript. We have substantially revised our manuscript by including new content and replying to address all Reviewer’s comments.

 Major concern:

  1. I really appreciate the two tables summarizing molecular mechanisms and potential therapeutic targets of PNI in PDAC. However, no schematic diagram was shown in this review. I highly recommended the authors to include schematic figures showing PNI related molecular mediators discussed in the manuscript and their functional implications in their associated pathways in PDAC. Besides, it’s better to label therapeutic targets with conventional and experimental agents on the schematic figure. The overlapping target proteins, signaling pathways, and functions between different molecular mediators without agents can be potential targets for further study, and this message should be delivered to the readers. The authors can refer to Li J, Kang R, Tang D. Cellular and molecular mechanisms of perineural invasion of pancreatic ductal adenocarcinoma[J]. Cancer Communications, 2021, 41(8): 642-660.

We have included three more figures as suggested, showing PNI-related signalling pathway and therapeutic targets (Figure 2), the potential role of extracellular vesicles in pancreatic cancer PNI (Figure 3), and a figure showing the therapeutics agents and relative targets (Figure 4) discussed in the text.

  1. Most of the main text are simple descriptions, lacking insight or summary. A good review should also be critical, summarize important research under the topic, identify consistencies and more importantly inconsistencies, and provide open questions for future research. For example, in the authors’ opinion, how should the field further explore the therapeutic targets of PNI in PDAC using single cell sequencing methods and other novel technologies?

We have substantially revised our review taking n to account this valuable suggestion and add more insight in our conclusion

  1. PNI is a common feature of different cancer types. Could the authors summarize PDAC-specific as well as broad-spectrum molecular mechanisms and potential therapeutic targets of PNI.

We have extended Section 2 to address this comment including a new subchapter (2.1. PNI Overview) and adding at the beginning of subsection 2.2 (PNI in pancreatic cancer) a broad introduction that explain the specificity of PNI in PDAC.

Minor concerns:

  1. The authors should spell out PNI (perineural invasion) when first mentioned in the abstract.

Corrected

  1. The authors should perform a thorough formatted and grammatical review for the manuscript and correct all typos. Eg, Line 105: “52,2% to 75,8%” should be “52.2% to 75.8%”

An extensive formatting and grammatical review have been performed as suggested

Reviewer 2 Report

This review focused on the perineural invasion (PNI) in pancreatic ductal adenocarcinoma (PDAC) from molecules and cellular signaling pathways towards drugs of Clinical Relevance. The authors introduced the importance of PNI in PDAC and stated the correlation between PNI and PDAC from five aspects including molecular and cellular signaling pathways, pain generation, extracellular vesicles, conventional treatments and cannabinoids in treatment. By analyzing these associations layer by layer, they indicated potential directions for PDAC treatment. In my perspective, the topic of this review makes sense, the content is substantial and the reference literature is relatively new. I recommend publication of this paper after some revisions.

1. In the introduction part, the majority focused on the risk and treatment of PDAC, a little part concerned on the neural influences on PDAC. The content can be adjusted slightly.

2. This review listed many signaling pathways of PNI in PDAC. Some schematic figures which showed the functions and relations of different signaling pathways of PNI in PDAC are recommended. These figures can make the article more readable.

3. Most of the treatment methods described in this review are included in targeted drug therapy. Is there any research on surgical treatment for PNI in PDAC?

Author Response

Response to Reviewer 2

Comments and Suggestions for Authors

This review focused on the perineural invasion (PNI) in pancreatic ductal adenocarcinoma (PDAC) from molecules and cellular signaling pathways towards drugs of Clinical Relevance. The authors introduced the importance of PNI in PDAC and stated the correlation between PNI and PDAC from five aspects including molecular and cellular signaling pathways, pain generation, extracellular vesicles, conventional treatments and cannabinoids in treatment. By analyzing these associations layer by layer, they indicated potential directions for PDAC treatment. In my perspective, the topic of this review makes sense, the content is substantial and the reference literature is relatively new. I recommend publication of this paper after some revisions.

We would like to take this opportunity to thank the Reviewer for his/her comments that have helped to improve our manuscript. We have substantially revised our manuscript by including new content and replying to address all Reviewer’ comments

  1. In the introduction part, the majority focused on the risk and treatment of PDAC, a little part concerned on the neural influences on PDAC. The content can be adjusted slightly.

The content of the Introduction chapter has been adjusted as suggested

  1. This review listed many signaling pathways of PNI in PDAC. Some schematic figures which showed the functions and relations of different signaling pathways of PNI in PDAC are recommended. These figures can make the article more readable.

We have included three more figures as suggested showing PNI-related signalling pathway and therapeutic targets (Figure 2), the potential role of extracellular vesicles in pancreatic cancer PNI (Figure 3), and a figure showing the therapeutics agents and relative targets (Figure 4) discussed in the text.

  1. Most of the treatment methods described in this review are included in targeted drug therapy. Is there any research on surgical treatment for PNI in PDAC?

We have included a new subchapter (6.1 Surgical treatment for PNI) that focuses on surgical treatment for PNI in PDAC